# Optimizing network propagation for multi-omics data integration

**Konstantina Charmpi**[1], **Manopriya Chokkalingam**[1], **Ronja Johnen**[1],
**Andreas Beyer**[1,2,3]*

1 CECAD Cologne Excellence Cluster on Cellular Stress Responses in Aging Associated Diseases, Cologne, Germany, 2 Center for Molecular Medicine Cologne (CMMC), Medical Faculty, University of Cologne, Cologne, Germany, 3 Institute for Genetics, Faculty of Mathematics and Natural Sciences, University of Cologne, Cologne, Germany

* andreas.beyer@uni-koeln.de

**Data Availability Statement:** Published data has been used and data sources are provided in the manuscript. Code is available from https://github.com/beyergroup/BioNetSmooth.

## Abstract

Network propagation refers to a class of algorithms that integrate information from input data across connected nodes in a given network. These algorithms have wide applications in systems biology, protein function prediction, inferring condition-specifically altered sub-networks, and prioritizing disease genes. Despite the popularity of network propagation, there is a lack of comparative analyses of different algorithms on real data and little guidance on how to select and parameterize the various algorithms. Here, we address this problem by analyzing different combinations of network normalization and propagation methods and by demonstrating schemes for the identification of optimal parameter settings on real proteome and transcriptome data. Our work highlights the risk of a 'topology bias' caused by the incorrect use of network normalization approaches. Capitalizing on the fact that network propagation is a regularization approach, we show that minimizing the bias-variance tradeoff can be utilized for selecting optimal parameters. The application to real multi-omics data demonstrated that optimal parameters could also be obtained by either maximizing the agreement between different omics layers (e.g. proteome and transcriptome) or by maximizing the consistency between biological replicates. Furthermore, we exemplified the utility and robustness of network propagation on multi-omics datasets for identifying ageing-associated genes in brain and liver tissues of rats and for elucidating molecular mechanisms underlying prostate cancer progression. Overall, this work compares different network propagation approaches and it presents strategies for how to use network propagation algorithms to optimally address a specific research question at hand.

## Author summary

Modern technologies enable the simultaneous measurement of tens of thousands of molecules in biological samples. Algorithms called network propagation or network smoothing are frequently used to integrate such data with already known molecular interaction data, such as protein and gene interaction networks. These methods distribute the information on molecular perturbations within the network and help identifying network regions that

**Funding:** KC received funding by the German Federal Ministry for Education and Research (BMBF, PhosphoNetPPM 031A259). MC received funding by the German Federal Ministry for Education and Research (BMBF, SyBACol 0315893). RJ received funding by the German Research Foundation (DFG, CRC 1310). The funders had no role in study design, data collection and analysis, decision to publish, or preparation of the manuscript.

**Competing interests:** The authors have declared that no competing interests exist.

are enriched for many perturbed (affected) molecules. Despite the popularity of these methods, there is a lack of guidance on how to optimally use them. Here, we highlight possible pitfalls when using incorrect network normalization methods. Further, we present different ways for optimizing the smoothing parameters used during network smoothing: the first approach maximizes the consistency between replicate measurements within a dataset; the second one maximizes the consistency between different types of 'omics' measurements, such as proteomics and transcriptomics. Using two multi-omics datasets, one from a cohort of prostate cancer patients, the other one from an ageing study on rat brain and liver tissues, we exemplify the effects of these strategies on real data.

## Introduction

Modern technologies allow us to measure many biomolecular properties at high-throughput, generating so-called 'omics data', such us genome-, transcriptome-, or proteome data. Whereas technical progress constantly improves the sensitivity and coverage of these methods, the analysis and interpretation of this data still suffers from technical noise and biological variation. Furthermore, the integration of data across 'omics layers' and/or across different individuals remains a challenge for computational biology. Network propagation (also called network smoothing) is a class of computational methods for addressing these problems by integrating such omics data with *a priori* known molecular relationships (e.g. protein-protein interaction maps). Thus, a particular strength of network propagation is the fact that prior knowledge is utilized for the analysis of new data, which potentially helps increasing the signal-to-noise ratio and which aids the mechanistic interpretation of results. Within the realm of molecular biology, network propagation has a wide range of applications such as imputation of missing values [1–3], protein function prediction [4], inferring condition-specifically altered sub-networks [5], and prioritization of disease genes [6,7].

Network propagation involves mapping the omics data onto pre-defined molecular or genetic networks that can be obtained from public databases such as STRING [8]. Subsequently, node scores are 'spread' to neighboring nodes through the edges in the network, i.e. node scores are updated using a weighted average of neighboring nodes. This averaging or smoothing of node scores will emphasize or 'amplify' regions in the network that are particularly associated with the phenotype, whereas regions with small changes around zero will be dampened. Various mathematical formulations of network propagation exist, such as Random Walk, Random Walk with Restart (RWR) [9], and Heat Diffusion (HD) [10]. Although the basic structure of these variants is common, they lead to mathematically distinct solutions, i.e. they are not just different names for the same method. Most methods perform some kind of pre-processing of the network (network normalization) and the way this network normalization is done determines the extent to which the network topology (i.e. the network structure) influences the final result of the algorithm. Further, the choice of the algorithm (e.g. RWR *versus* HD) influences the 'information sharing' between connected nodes. Finally, tuning parameters that determine the magnitude of spreading to adjacent nodes (spreading coefficients) have to be chosen. Depending on the choice of these tuning parameters, the final node scores can either be close to the initial values before network propagation or be strongly influenced by direct and distant neighbors in the network. Whereas some combinations of network normalization and network propagation methods can be disfavored due to formal reasons, the optimal choice of methods and parameters is often dependent on the research question and the available data. Despite the recent popularity of network propagation and many published

examples, there is incomplete understanding about how the choice of the network normalization method, the network propagation algorithm and the tuning parameters influence the result, and there is a need for guidance on how to optimize the analysis scheme.

In order to address these questions, we focus on two of the most popular network propagation algorithms: RWR and HD. For the algorithm description, we assume that the vector $F_0$ contains the initial node scores that are to be propagated, such as fold changes of transcripts or phenotypes measured per gene. These node scores are 'mapped' onto a pre-defined network, which may or may not be weighted. This network is represented by a (commonly normalized) matrix $W$.

Random Walk on (unweighted) graphs is an iterative walker that starts from some node and at each time step transitions randomly to one of the neighboring nodes of the current node on the graph. In a weighted network, the transition to adjacent nodes is proportional to the edge weights. The 'time' that the walker spends on each node will depend on the 'sharing' of information with that node. A problem of Random Walk is that it does not converge to a solution that reflects the input data (i.e. the vector $F_0$). Instead, the final solution will be solely dependent on the network topology. Random Walk with Restart is different in this respect: it is a variant of Random Walk with an option to transition back to the starting node at every iteration with the restart probability $(1 - \alpha)$ $(0 \leq \alpha \leq 1)$. In the network propagation framework the restart probability determines the magnitude of signal that has to be retained at the query nodes, i.e. it can be interpreted as a dampening factor on long walks. Thus, a small $\alpha$ will keep the node scores close to the initial scores, whereas a large $\alpha$ will average scores of connected nodes in the network more strongly. RWR is an iterative procedure, the node scores are propagated on the network until they converge [11]—assuming that the convergence conditions are satisfied. Node scores at the $i^{th}$ iteration are updated according to:

$$F_i = (1 - \alpha)F_0 - \alpha W F_{i-1}, (i = 1, 2, \ldots) \tag{1}$$

where $\alpha$ is the spreading coefficient representing the fraction of spread, $W$ is a matrix representing the normalized network, $F_0$ is a vector containing the initial node scores and $F_i$ is the vector representing the propagated scores. The convergence is estimated through calculating the norm of $F_i$—$F_{i-1}$ and the iteration is stopped when the value falls below a certain threshold (e.g. below $10^{-6}$) [5]. If $F^a$ denotes the limit of the sequence $\{F_i\}$ then

$$F^a = (1 - \alpha)(I + \alpha W)^{-1} F_0,$$

where $I$ is the identity matrix.

Heat Diffusion (HD; also called diffusion kernel) is a continuous-time analogue of RWR and models a fluid flow over a fixed number of time steps. These are defined by a time parameter $t$, which controls the spreading of signal [6]. The amount of 'fluid' that ends up at all network nodes can be computed according to

$$F_t = exp^{-Wt} F_0, \tag{2}$$

where $t$ in $[0, \infty)$ is the time diffusion and is a tunable parameter, and $F_t$ is the vector representing the propagated node scores. When $t$ is set to zero, the node scores are not propagated to adjacent nodes (i.e. $F_t = F_0$) and when $t$ approaches infinity the solution will exclusively depend on $W$, i.e. the network topology, and information on the initial node scores is lost [12].

In this study we have systematically analyzed the performance of these two algorithms using *Rattus norvegicus* ageing mRNA and protein abundance data from two different metabolically active tissues: brain and liver at two age groups [13] as well as mRNA and protein data from prostate cancer patients of different grade groups [14]. We focused on two

questions: (1) how does the graph normalization (i.e. the computation of $W$) determine the influence of network topology on the final result? (2) Which strategies can be used for finding optimal choices for the smoothing parameters $\alpha$ and $t$?

Firstly, we are demonstrating that an inappropriately chosen graph normalization method can lead to a topology bias, i.e. a biased increase or reduction of node scores that is exclusively due to the network structure and independent of the input vector $F_0$. Secondly, we used the mean squared error and its decomposition into bias and variance as a first criterion to select optimal parameters. Thirdly, transcriptome replicates were utilized to illustrate the effect of changing the spreading parameters in order to increase the consistency between replicate measurements. Fourthly, we employ RWR and HD to increase the consistency between transcriptomics and proteomics data. These analyses underlined again the differences and similarities between the methods and emphasized the importance of choosing optimal spreading parameters. Finally, we used network propagation to impute missing values and uncover ageing-associated genes as well as to identify sub-networks distinguishing prostate tumors of different grades.

## Results

### Graph normalization and topology bias

The adjacency matrix $A$ of a network represents the direct node-to-node relationships (see *Material and Methods*). In other words, the square matrix $A$ contains non-zero entries $a_{ij}$ for all node pairs $i$ and $j$ that are connected in the network. The non-zero entries $a_{ij}$ correspond to edge weights (or 1 if the network is unweighted). In case of undirected networks $A$ is symmetric. Network propagation algorithms do not usually operate directly on $A$ (i.e. setting $W = -A$), because this would strongly bias the results in favor of nodes with many neighbors and because convergence cannot be guaranteed for all values of the smoothing parameter. Different graph normalization methods have been proposed, here we focus on three popular forms: the Laplacian transformation, the normalized Laplacian, and the degree normalized adjacency matrix (Table 1). The Laplacian is defined by

$$W_L = D - A, \tag{3}$$

where $D$ is a diagonal matrix containing the node degrees such that the entry $d_{ii}$ is the sum of the $i$-th row of $A$. The eigenvalues of the matrix $W_L$ are not upper bound by 1, and therefore RWR is not guaranteed to converge for all $0 \leq \alpha \leq 1$ [15,16]. Thus, this matrix is only used for HD and not further considered for RWR.

The Laplacian $W_L$ accounts for node degrees only in its diagonal, whereas the actual edge weights (off-diagonal elements) remain unchanged. The normalized Laplacian $W_{\hat{L}}$ also normalizes off-diagonal elements using the degrees of the interacting nodes $i$ and $j$:

$$W_{\hat{L}} = D^{-\frac{1}{2}} W_L D^{-\frac{1}{2}}. \tag{4}$$

**Table 1. Different graph normalization approaches and their impact on propagated scores.** Network propagation leads to topology bias when the normalized Laplacian of the graph is used, whereas the degree row-normalized adjacency matrix does not lead to bias on the hub nodes. The Laplacian of the graph cannot be used for RWR because the iterative process is not guaranteed to converge for all $\alpha$'s. **Yes**: presence of topology bias, **No**: absence of topology bias for the respective combination of propagation algorithm and graph normalization approach. The symbol "-" indicates that convergence is not guaranteed for all values of the smoothing parameter.

| Method | Laplacian: $W_L$ | Normalized Laplacian: $W_{\hat{L}}$ for HD and $W_{\hat{L}_D}$ for RWR | Degree row-normalized adjacency: $W_{\hat{D}}$ for HD and $W_D$ for RWR |
|---|---|---|---|
| HD | No | Yes | No |
| RWR | - | Yes | No |

For RWR, we additionally set the diagonal of $W_{\hat{L}}$ to zero (see *Material and Methods*):

$$W_{\hat{L}D} = W_{\hat{L}} - I,$$

where $I$ is the identity matrix.

Whereas the Laplacian is based on subtracting the degree matrix $D$, the degree row-normalized adjacency matrix is obtained by multiplying the inverse of $D$ with $A$. We normalize the matrix row-wise [17–19] and define the degree row-normalized adjacency matrix as

$$W_D = -D^{-1}A. \tag{5}$$

The use of the degree row-normalized adjacency matrix with HD leads to an exponential increase of the propagated node scores by the factor of $e^t$. To avoid that for very large $t$ the algorithm may run into numerical instabilities, the identity matrix $I$ was added to the degree row-normalized adjacency matrix to ensure that the row sums are equal to zero:

$$W_{\hat{D}} = W_D + I. \tag{6}$$

The adjacency matrix can also be normalized column-wise [6].

To illustrate the effect of different graph normalization methods, we used an undirected network for the rat *Rattus norvegicus*, because the same network was subsequently used to investigate the effect of the smoothing parameters. We retrieved the network from the protein interaction database STRING version 10.5 [8] and filtered for high confidence edges (probability of an interaction $\geq 0.9$) and the largest connected network component (9,388 nodes). However, the conclusions in this section regarding the choice of $W$ are independent of any particular network.

The core idea of network propagation is that regions in a network enriched with high-scoring nodes get 'amplified', whereas regions with many scores close to or varying around zero get 'dampened'. Thus, the result should depend on both, the structure of the network (its topology) and the initial non-random distribution of node scores. The network topology alone, i.e. independent of the node scores, should not lead to increased node scores after network propagation. However, inappropriately chosen graph normalization methods can lead to a topology bias. To demonstrate this effect, we computed for an undirected network the propagated results for an input vector of ones, i.e. setting all entries in $F_0$ to 1. Without a topology bias all nodes should have identical scores after network propagation, because there was no initial 'clustering' of high-scoring nodes in any sub-network. Note, that this is equivalent to the expected outcome for an infinite number of random input vectors with mean 1.

The normalized Laplacian leads to a topology bias for both RWR and HD, i.e. hub nodes get higher scores than non-hub nodes (Fig 1). However, using either the Laplacian (HD only) or the row-normalized adjacency matrix (HD & RWR) prevents a topology bias, i.e. after propagating the unit vector all nodes have identical scores (see *Material and Methods*; Table 1). Since the column-normalized adjacency matrix [6] also leads to topology bias (the limiting vector being proportional to the vector of node degrees for instance for RWR and $\alpha = 1$), we decided to use the row-normalized adjacency matrix in this analysis. All subsequent analyses employed the degree row-normalized adjacency matrix in order to avoid any topology bias. With this choice of graph normalization, each input vector will converge to a vector with the same value for all elements as $\alpha$ approaches one and $t$ goes to infinity–in other words there will be no variability between the nodes for $\alpha = 1$ and $t$ large enough. For the sake of simplicity we have decided to use this normalization method for the remaining parts of this publication. However, we do not intend to imply that the degree row-normalized adjacency matrix would be the sole possible choice. In fact, many other graph normalization methods have been

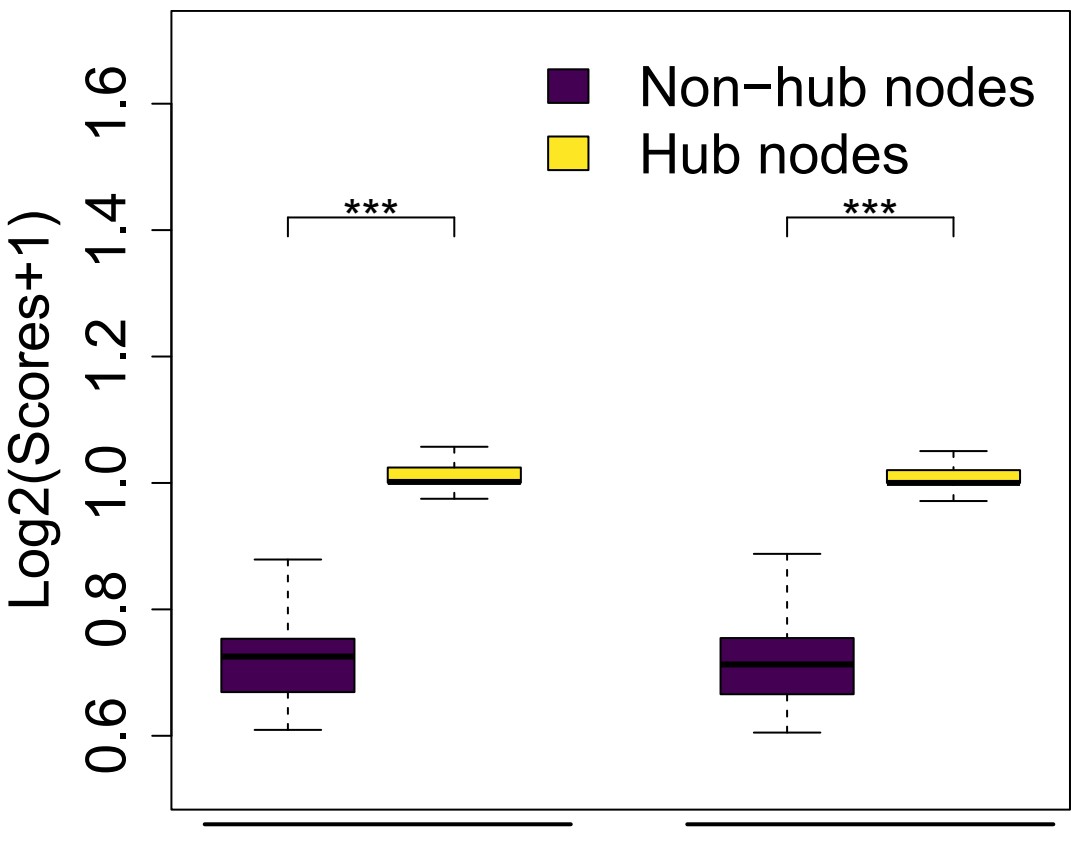

**Fig 1. Normalized Laplacian induces a topology bias.** Distributions of log-transformed node scores after network propagation using the normalized Laplacian ($W_{\hat{L}}$ and $t = 0.7$ for Heat Diffusion, and $W_{\hat{L}D}$ and $\alpha = 0.5$ for Random Walk with Restart). The input vector was a unit vector, i.e. all nodes had identical initial scores. Hub nodes (top 10% nodes with the highest degree) gain higher average scores, whereas non-hub nodes (bottom 10% nodes with the lowest degree) get lower average scores (Two-sided Wilcoxon rank sum test p-value $< 2.2 \cdot 10^{-16}$ (***)).

proposed that might be suitable for a specific application of network propagation [11,20]. More details regarding the presence or absence of a topology bias using different graph normalization and propagation approaches are described in the section *Material and Methods*.

## Using network propagation to maximize within-dataset consistency

Spreading coefficients are tunable parameters of HD and RWR, and they control the amount and distance of signal spread on the interaction network. For RWR, the parameter $\alpha$ defines the fraction of node scores that has to be propagated to adjacent nodes at each iteration and it ranges between 0 and 1. Likewise in HD, the parameter $t$ corresponds to the fraction of diffusion and it ranges from 0 to infinity. In existing publications it is sometimes unclear to what extent different spreading parameters were tested and based on which criteria final parameters were selected. Therefore, we wanted to know to what extent final results depend on the chosen spreading parameter and we wanted to test optimization criteria for selecting them. If the

network topology is informative of gene-gene relationships, i.e. if it relates to the covariance structure in the data, one would expect that network propagation would drive noisy measurements closer to their (true) population means.

Here, we demonstrate that this notion can be used to find optimal choices for the spreading parameters. If network propagation enhances true biological signal in the data it should improve the consistency between replicate measurements. However, too excessive 'smoothing' might yield node scores that become similar between replicates, but which are independent of the input data and solely depend on the network structure. We demonstrate the utility of this idea by comparing individual replicate measurements to the mean of all replicates (before propagation), thus assuming that the average over all replicates is closer to the true population mean.

The spreading parameters determine how close the final solution will be to the initial vector $F_0$ *versus* all converging to a common value. Thus, it has been suggested that network propagation can be understood as a regularization approach [21]. Here we show that network propagation yields a bias-variance tradeoff, using the following intuition: smoothing/'averaging' the data tends to make it less noisy, but this information sharing also creates biases (on the mean values). In other words, the propagated scores/values (e.g. $\log_2$ fold changes) of the genes will vary less across the replicates/samples, but will be distributed around a mean (propagated mean) which is different from the true mean. More rigorously, let $F_0 = (F_{01}, F_{02}, \ldots, F_{0p})^T$ be the random vector of $p$ scores (e.g. fold changes) with population mean vector $\mu$. Using the propagated vector $MF_0$ as estimate for $\mu$ (where $M = exp^{-Wt}$ or $(1-\alpha)(I+\alpha W)^{-1}$) with mean vector $M\mu$, we can compute the mean squared error (MSE) of our estimate for gene $i$ as $\mathbb{E}[((MF_0)_i - \mu_i)^2]$, where $\mathbb{E}[\cdot]$ denotes the expectation and write

$$\mathbb{E}[((MF_0)_i - \mu_i)^2] = \mathbb{E}[((MF_0)_i - (M\mu)_i + (M\mu)_i - \mu_i)^2]$$
$$= \mathbb{E}[((MF_0)_i - (M\mu)_i)^2] + \mathbb{E}[((M\mu)_i - \mu_i)^2] + 2\mathbb{E}[((MF_0)_i - (M\mu)_i)((M\mu)_i - \mu_i)]. \quad (7)$$

This formula corresponds to the classical bias-variance decomposition of MSE- where the first term corresponds to the variance, the second to the bias$^2$ and the third (error term) is equal to zero since $\mathbb{E}[(MF_0)_i] = (M\mu)_i$. Computing an average MSE over the genes we get

$$\sum_{i=1}^{p} \mathbb{E}[((MF_0)_i - \mu_i)^2]/p = \sum_{i=1}^{p} \mathbb{E}[((MF_0)_i - (M\mu)_i)^2]/p + \sum_{i=1}^{p}((M\mu)_i - \mu_i)^2/p, \quad (8)$$

where $p$ is the number of nodes (i.e. genes or proteins), the first term corresponds to an average variance and the second to an average bias$^2$. When $\alpha = t = 0$, the matrix $M$ is the identity matrix so the average bias$^2$ is equal to zero, but is expected to increase with increasing spreading parameters since the propagation (with the row-normalized adjacency matrix) eventually 'forces' all genes to a common value. On the other hand, the average variance between replicates is expected to decrease with more aggressive smoothing since smoothed node scores will assume a weighted average ($MF_0$) using information from more and more genes (i.e. the sample size is increased and the noise terms tend to cancel out each other). Here, we propose to choose $\alpha$ or $t$ such that they minimize the average MSE, i.e. minimizing the sum of bias$^2$ and variance.

To test this idea we used published transcriptome data of rat brain and liver tissues at 6 months ('young') and 24 months ('old') of age (three replicates each) [13]. After pre-processing the data with standard packages, we computed (average) $\log_2$ fold changes per tissue comparing old *versus* young rat tissues ($\overline{old}$ *versus* $\overline{young}$; where $\overline{old}$ and $\overline{young}$ denotes the averages of the respective replicates) using DESeq2 [22]. In order to evaluate the variation between replicate measurements, we computed fold changes of each single old sample against the average

of all young samples, yielding three fold change estimates for each gene ($old_1$ versus $\overline{young}$, $old_2$ versus $\overline{young}$, $old_3$ versus $\overline{young}$). Likewise, we computed a fold change for each young sample versus the average of all old samples, creating another set of three fold change estimates per gene ($\overline{old}$ versus $young_1$, $\overline{old}$ versus $young_2$, $\overline{old}$ versus $young_3$). Thus, this scheme results in exactly one fold-change estimate per sample. We refrained from directly comparing expression levels (i.e. not computing fold changes) for two reasons: first of all, comparing absolute expression levels would be dominated by highly expressed genes. Second, estimating fold changes is more relevant for the majority of research questions.

First, we confirmed empirically that increasingly aggressive network propagation reduces the within-replicate variance (node scores become more similar) and increases the between-gene correlation (see S3 Fig).

Next, we asked if network propagation could help to denoise the data. Since we do not know the ground truth fold changes, we used the following notion: we assumed that the fold changes of the averaged replicate measurements are closer to the true fold changes than fold changes resulting from the individual replicates. Hence, we figured if network propagation performed on individual replicates'moves' fold changes closer to the across-replicate average (before network propagation), this would reflect a noise reduction. In order to test this notion, we performed network propagation on the $log_2$ fold changes of each replicate/tumor sample as well as the average $log_2$ fold changes with varying spreading parameters $\alpha$ and $t$.

Here we estimated the expectation $\mathbb{E}[((MF_0)_i - \mu_i)^2]$ by the replicate value $((MF_0^k)_i - \mu_i)^2$ where $F_0^k$ is the vector of fold changes of replicate/sample $k$ and the mean vector $\mu$ by the initial average $log_2$ fold changes. Using this we computed the average MSE for each sample in each dataset with varying spreading parameters for HD and RWR. Likewise, we estimated the variance from Eq (8) per sample, approximating the expectation $\mathbb{E}[((MF_0)_i - (M\mu)_i)^2]$ with the corresponding replicate value $((MF_0^k)_i - (M\mu)_i)^2$, and the average $bias^2$ by computing the squared difference between the propagated average $log_2$ fold changes and the initial average $log_2$ fold changes.

Network propagation could reduce the MSE for the rat brain data (Fig 2A and 2B), whereas it did not reduce the MSE of the liver data (Fig 2C and 2D). We noted that the initial absolute fold changes in the liver were on average much larger than in the brain (S4 Fig), indicating that upfront there was a stronger signal in the liver. Hence, it is possible that the relatively strong signal in the liver data could not be further improved through network propagation (see Discussion for more details). To further test this hypothesis, we added noise to the liver data. The noise added to each gene was drawn from a normal distribution with zero mean and (a gene-specific) variance equal to the estimated inter-replicate variance of that gene (see *Material and Methods* for more details). We quantified the ability of network propagation to recover the original expression change values by computing the MSE with respect to the original mean before adding the noise. This analysis confirmed that network propagation successfully denoised that data, i.e. on average node scores were brought closer to their original mean (Fig 2E and 2F).

To consolidate the idea that minimizing the MSE could be used as an optimization criterion for the spreading parameters, we also tested a dataset of 66 prostate cancer (PCa) samples belonging to four different grade groups (G1, G2, G3, G4/5) of increasing aggressiveness and 39 patient-matched benign samples from 39 patients [14]. We quantified mRNA and protein $log_2$ fold changes for the tumor samples (*versus* benign) as described in Charmpi *et al*. [14] as well as average (mRNA and protein) $log_2$ fold changes across the tumor samples. In the PCa cohort, moderate smoothing minimized the MSE on the mRNA layer, while more aggressive smoothing was required for protein data (Fig 2G–2J). Next, we performed the same analysis

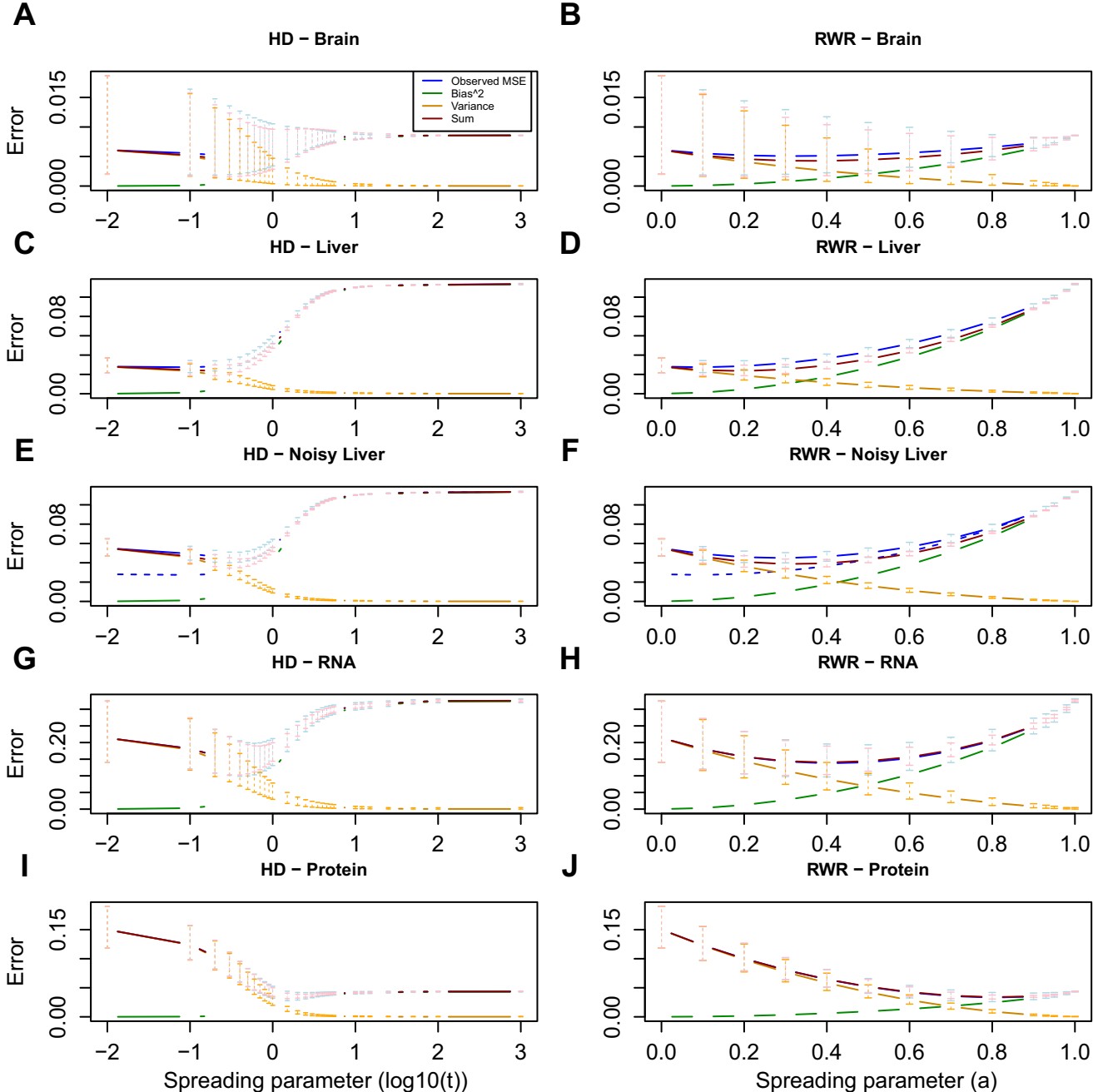

**Fig 2. Bias-variance decomposition of the Mean Squared Error (MSE).** Mean squared error curve (blue, averaged across samples) across the values of the spreading parameter for the brain (**A**, **B**) and liver (**C**, **D**) and for the liver data with added noise (**E**, **F**) in Ori *et al.* [13] dataset, and for the mRNA (**G**, **H**) and protein layer (**I**, **J**) in the PCa dataset using HD (**A**, **C**, **E**, **G**, **I**) and RWR (**B**, **D**, **F**, **H**, **J**). Minimum MSEs are circled (blue). Additionally, the decomposition is depicted: the bias$^2$ curve (green), the variance curve (orange) (as defined in *Material and Methods*) and their sum (red) are shown. The sum (red curve) approximates very well the observed MSE (blue curve). For the quantities with replicate values (i.e. observed MSE, variance and sum), error bars have been added to represent the actual distribution at the respective spreading parameters. The replicate data is firstly log-transformed. Subsequently, the average and standard deviation (SD) across the six replicates is computed for each value of the spreading parameter in the transformed space. Finally, the average, the lower error bound (average-SD) and the upper error bound (average+SD) in the log-transformed space are transformed back to the original space (i.e. the points correspond to the geometric mean). In (**E**, **F**), the dashed curve (blue) corresponds to the MSE curve of the original liver data (i.e. the (blue) curve depicted in (**C**) and (**D**) respectively).

individually for each tumor grade group, as opposed to combining all tumors in one analysis. The amount of network smoothing required to minimize the MSEs in this case was almost the same as for the global analysis (S5 Fig).

To investigate which genes benefit the most from network propagation, we also computed the MSE for each gene separately in the PCa cohort with RWR. We observed a high variability of the MSEs at $\alpha = 1$ for the mRNA as opposed to the protein layer suggesting that the mean values of the genes at the mRNA layer are more variable compared to the protein layer (Fig 3A and 3B). We would expect that genes that are highly variable between replicates benefit most from network propagation. To quantify the propagation gain, we computed for each gene the difference between its MSE at $\alpha = 0$ (i.e. without network propagation) and its MSE at $\alpha = 0.4$ (global optimum; Fig 2H). We also computed for each gene its mean $\log_2$ fold change across the tumor samples, its variance and the corresponding ratio (mean/standard deviation). All three quantities correlated with the MSE difference, i.e. the propagation gain (Fig 3C, 3D and 3E). Regarding the network topology, we expected genes with an 'informative' neighborhood (i.e. neighbors with similar mean $\log_2$ fold changes) to benefit more from propagation compared to genes in a random neighborhood or with opposite-sign mean $\log_2$ fold change neighbors. To quantify the neighborhood 'information', we computed for each gene the mean $\log_2$ fold change of its neighbors. We observed a weak, but significant correlation between the genes' $\log_2$ fold changes and the averaged fold changes of their neighbors (Fig 3F), confirming that the network is informative on this dataset. Further, we confirmed that indeed genes with informative neighbors achieved lower MSEs after network propagation compared to other genes (Fig 3G). Thus, this analysis confirms the intuition that genes with informative neighbors benefit most from network propagation.

Subsequently, we tested an alternative way to quantify the inter-replicate consistency by using the correlation between replicate fold change estimates and average fold changes. The average inter-replicate fold change correlations of the rat transcriptomes before smoothing were 0.693 and 0.877 for brain and liver, respectively. In the case of the brain samples, network propagation could improve the inter-replicate consistency at least when using HD (Fig 4A and 4B). However, for the liver the already high consistency could only marginally be increased with network propagation (Fig 4C and 4D)–an observation that was consistent with the MSE-based analysis. Importantly, more aggressive smoothing with $\alpha$ and $t$ greater than the optimum first reduced the inter-replicate consistency, suggesting that biological signal was diminished. Even further smoothing then lead to again larger correlation scores. However, this 'over smoothing' yields node scores that will eventually converge against a common value (see also *Material and Methods*). An important feature of the analysis scheme presented here is that it provides guidance on how to prevent 'over smoothing', i.e. spreading parameters should be selected either at the minimum of the bias-variance tradeoff (i.e. minimizing the MSE) or well before the first dip of the inter-replicate correlation. Importantly, the optimal smoothing parameters resulting from the two criteria were different. Thus, the choice of the optimization criterion will influence the outcome.

Some of the patients in the PCa cohort (27 out of 39) had two tumor regions analyzed (TA1 and TA2). Although prostate cancer tumors can be spatially heterogeneous, we had previously shown that the intra-patient variation is on average smaller than inter-patient variation [14]. Hence, within limits, the two samples from the same tumor can be treated like replicate measurements. Thus, we tested if network propagation would further increase the intra-patient similarity. For that, we computed the correlation between the propagated mRNA $\log_2$ fold changes of TA1 and corresponding propagated mRNA $\log_2$ fold changes of paired TA2 for each such patient with the two propagation algorithms for varying spreading parameters. For most patients, the intra-patient similarity increased with network smoothing (Fig 4E and 4F).

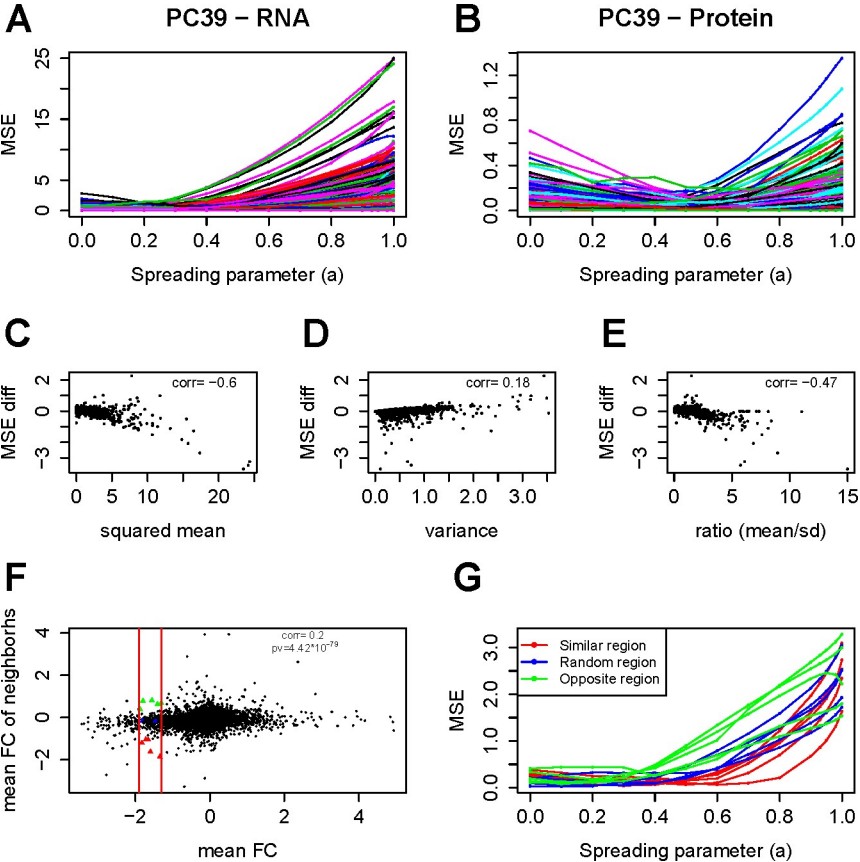

**Fig 3. Factors affecting network propagation gain of individual genes. A, B:** MSE curves of individual genes for the mRNA (**A**) and protein (**B**) layer of the PCa study using RWR. **C, D, E:** Impact of network propagation on gene-specific MSEs. The panels show the change of the MSE (MSE difference comparing value at $\alpha = 0$ and $\alpha = 0.4$) *versus* the corresponding (squared) mean $\log_2$ fold changes (**C**), *versus* the inter-replicate variance (**D**), and *versus* the corresponding ratios (mean in absolute/SD) (**E**) for the mRNA layer. The correlation coefficient is given in all three panels. **F:** Average $\log_2$ fold change of the genes' neighbors *versus* their own fold changes for the mRNA layer. The correlation coefficient and corresponding p-value are given. The two vertical red lines have abscissa (-1.9) and (-1.3). Colored points within this area have been selected to generate MSE curves in (**G**). **G:** MSE curves of the colored points in (**F**). The genes were selected to have similar mean $\log_2$ fold changes to eliminate the effect of the mean value. Red curves correspond to genes with similar mean value neighbors, blue correspond to genes in a random neighborhood and green to genes with opposite sign mean value neighbors. Genes with informative neighbors (red curves) achieved lower MSEs after network propagation compared to other genes (blue and green curves).

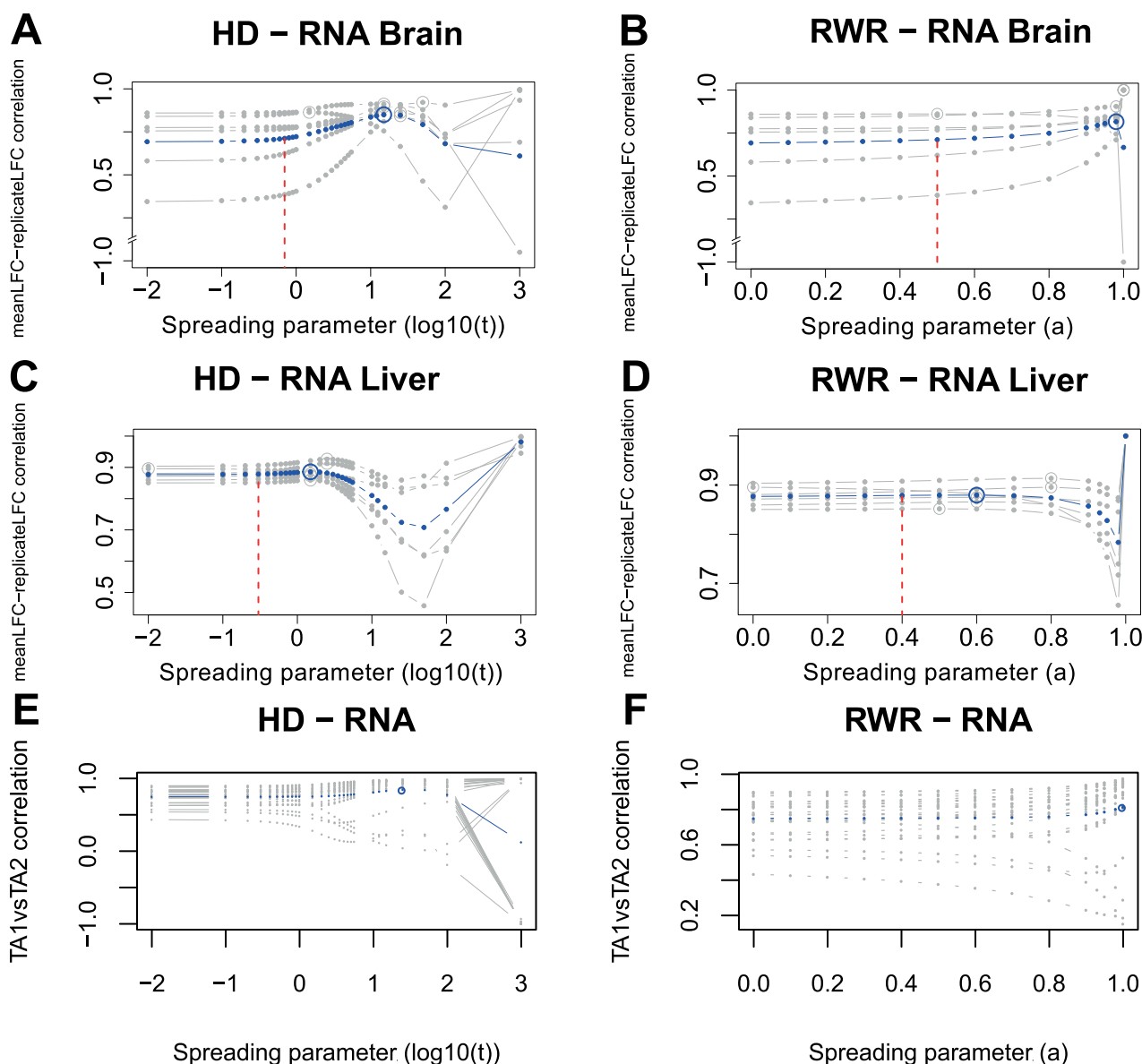

**Fig 4. Inter-replicate consistency and within-patient similarity across network smoothing.** Correlation of replicate-wise propagated $\log_2$ fold changes with propagated average $\log_2$ fold changes of the transcriptome for the brain (**A, B**) and liver (**C, D**) using HD (**A, C**) and RWR (**B, D**). Gray lines are correlations of individual replicates with the average fold changes obtained by combining all replicates. Blue lines are averaged correlations across replicates at the respective spreading parameters for HD and RWR. Maximum correlations are circled (gray and blue). The vertical dashed red lines denote the optimal spreading parameters from the between-dataset analysis (i.e. comparing mRNA and protein propagated scores; see Fig 5). **E, F:** Correlations between propagated mRNA $\log_2$ fold changes of TA1 and corresponding propagated mRNA $\log_2$ fold changes of paired TA2 across the values of the spreading parameter for the 25 PCa patients with HD (**E**) and RWR (**F**). Each gray curve corresponds to a patient while the average curve is depicted in blue. Average maximal correlations are circled. For computing the correlation, only measured network nodes within each dataset were used.

In the case of four patients it decreased with increasing network propagation. However, those cases corresponded mostly to tumors with already low initial intra-patient similarity (Fig 4E and 4F). Hence, these patients might correspond to cases with two independent tumors (clones) [14]. Although these observations were supported by both propagation algorithms, a local maximum for the average within-patient similarity was attained only with HD-

suggesting that HD might be more appropriate for this dataset. Using the Spearman rank correlation (which is independent of the specific distribution of the data and less affected by outliers) instead of the Pearson correlation does not change those conclusions (S6 Fig).

## Network propagation improves between-dataset consistency

As yet another optimization criterion, we used the correlation between mRNA and matching proteomics data. Although protein levels are determined by factors that are independent of their coding mRNA levels (such as variable translation rates or protein turnover), one typically observes a significant correlation between steady-state mRNA fold changes and the respective protein fold changes [23]. To test the utility of this criterion, we additionally used protein data from the Ori *et al.* [13] study. In this case, we used average $\log_2$ fold changes between old and young rats in the two layers. Protein and mRNA levels were available for 1,772 common genes in brain and 1,670 in liver that were used for the subsequent analyses. Without network propagation we observed small but significant correlations between mRNA and protein $\log_2$ fold changes in both, brain and liver (brain: corr = 0.236, $p < 2.2 \cdot 10^{-16}$; liver: corr = 0.308, $p < 2.2 \cdot 10^{-16}$).

Network propagation slightly improved the protein-mRNA correlations in the brain (HD_Cor = 0.244 at $t$ = 0.7, RWR_Cor = 0.245 at $\alpha$ = 0.5; Fig 5A and 5B). Here, network propagation was conducted independently on the mRNA and protein data. Thus, it would be possible to use different spreading parameters for the two types of data. However, using different spreading parameters for the two data types did not further increase the protein-mRNA correlation (S2 Fig). In the case of the liver data network propagation did not further increase the protein-mRNA correlation (Fig 5C and 5D). This lack of improvement is consistent with the inter-replicate analysis above, which showed that the signal-to-noise ratio was much larger in the liver compared to the brain samples (S4 Fig). Also the initial protein-mRNA correlation was higher in the liver compared to the brain, suggesting that network propagation had a greater potential of removing noise from the brain data compared to the liver data.

Next, we computed the correlation between the propagated mRNA and protein $\log_2$ fold changes of each PCa tumor sample with the two propagation algorithms for varying spreading parameters. For most of the samples, the between-layer similarity increased with smoothing (Fig 5E and 5F). The few cases where the network propagation did not improve the correlation corresponded mostly to tumor samples with low initial correlations (Fig 5E and 5F). Despite the differences in the obtained curves, these observations were supported by both propagation algorithms. However, with HD a local maximum for the average between-layer similarity was attained suggesting that HD might be more informative (as before for the within-patient similarity) for this dataset. In addition, we used Spearman correlation to measure the between-dataset consistency in the ageing study and the prostate cancer study. The conclusions are in agreement with those from Pearson correlation (S7 Fig).

Importantly, this analysis again exposed the risk of 'over smoothing', i.e. too aggressive network propagation leads to equal scores between the nodes both at the mRNA and protein layer (Fig 5, *Material and Methods*).

## Functional interpretation of network propagation results

One possible application of network propagation is the imputation of missing values [2]. For example, shotgun proteomics measurements often do not quantify all proteins in a sample. One might use network propagation to impute those missing values by utilizing measured protein levels of neighboring proteins in the network. In order to test this idea and in order to further validate the plausibility of network propagation results, we imputed expression fold

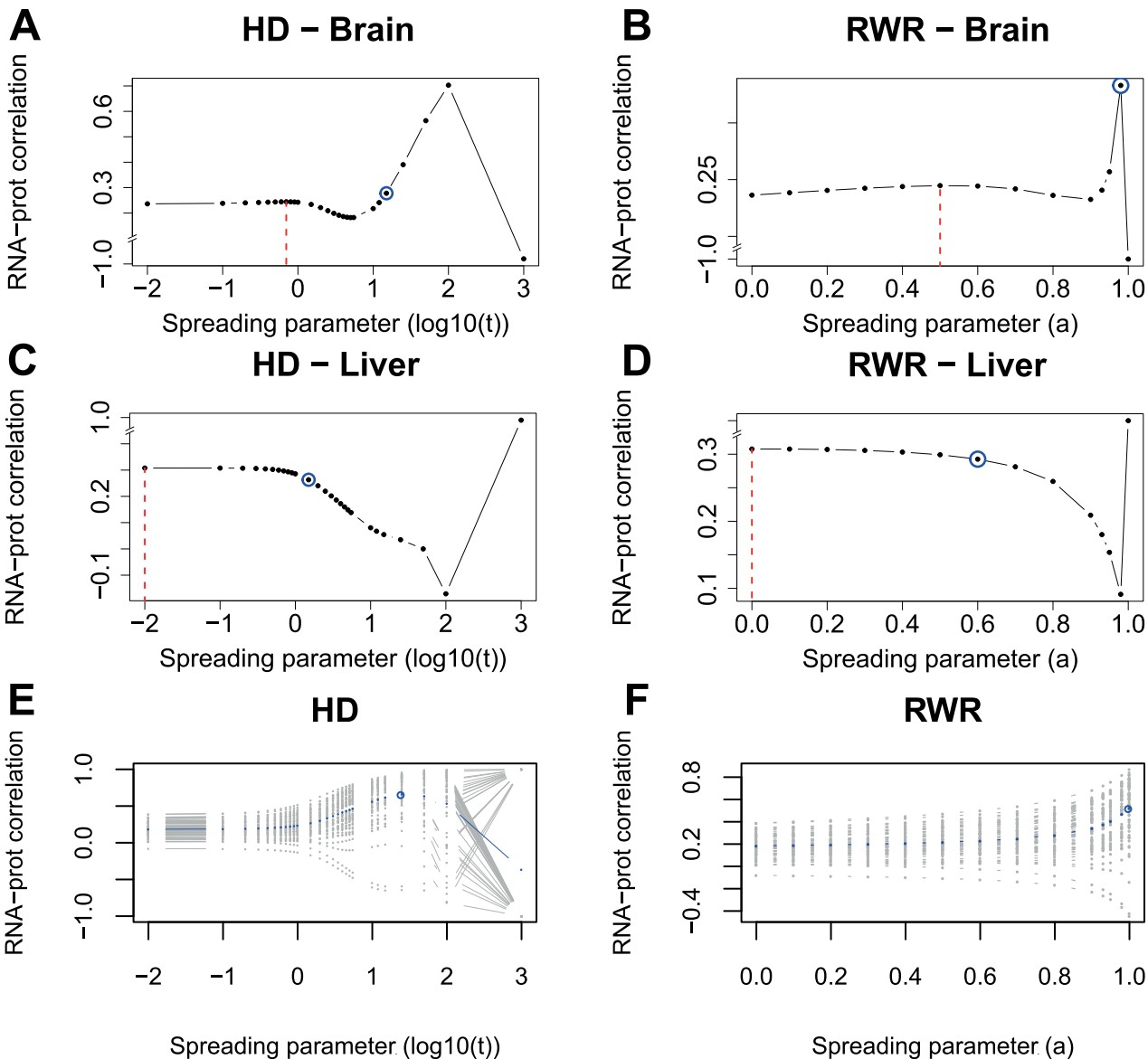

**Fig 5. Network propagation improves the correlation between mRNA and protein levels of ageing tissues (brain and liver) and PCa samples. A, B:** correlations between propagated scores from mRNA $\log_2$ fold changes and protein weighted mean $\log_2$ fold changes from old brain samples with varying spreading coefficients for HD (**A**) and RWR (**B**). **C, D:** Correlations of propagated scores from mRNA and protein $\log_2$ fold changes of liver samples with HD (**C**) and RWR (**D**). Dashed red lines: local maximal correlations from the between-dataset consistency analysis. Blue circles: average maximal correlations from the within-dataset consistency analysis. **E, F:** Correlations between propagated mRNA $\log_2$ fold changes and corresponding propagated protein $\log_2$ fold changes across the values of the spreading parameter for the 63 PCa tumor samples with HD (**E**) and RWR (**F**). Each gray curve corresponds to a tumor sample while the 'average' curve is depicted in blue. Average maximal correlations are circled. Correlations were calculated using genes that were expressed and quantified with both RNA-Sequencing and MS proteomics as well as present in the functional interaction network (n = 1,772 genes for brain, n = 1,670 for liver, n = 1,828 for PCa).

changes (young versus old) for missing proteins and mRNAs with the goal to recover known ageing-associated proteins and transcripts. We set initial protein $\log_2$ fold changes of 7,149 missing proteins for brain and 7,318 for liver to 0, while using observed $\log_2$ fold changes (old *versus* young) for all other 2,239 proteins for brain and 2,070 for liver. Then, we performed network propagation on the $\log_2$ fold changes, assuming that unobserved proteins in network regions with large $\log_2$ fold changes will likely also be strongly affected by ageing. Likewise, we

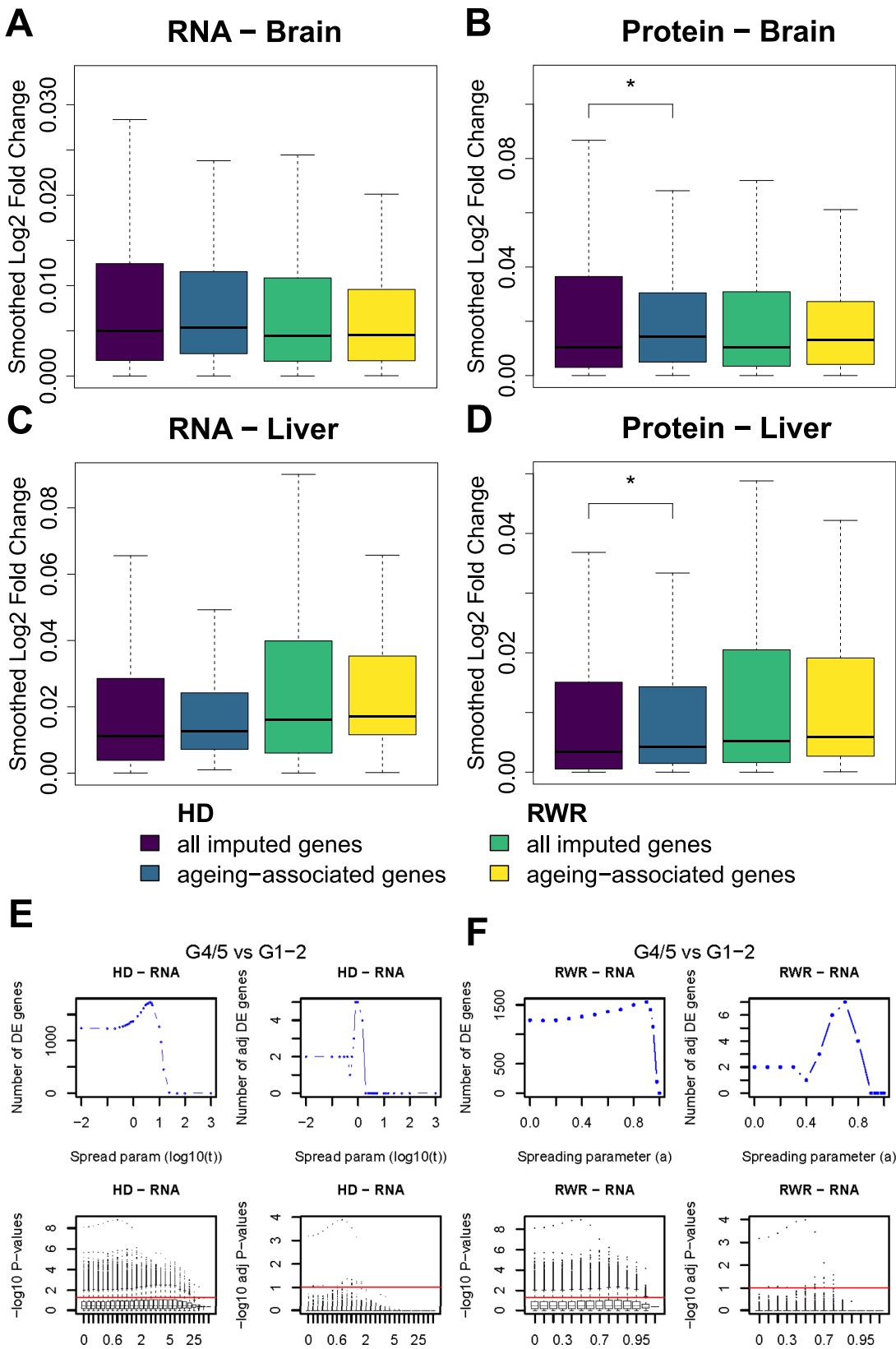

**Fig 6. Network propagation identifies ageing-associated genes as well as genes distinguishing prostate tumors of different grades.** RNA (**A** and **C**) and proteome (**B** and **D**) $\log_2$ fold changes were smoothed on the network using RWR (yellow/green colored) and HD (purple/blue colored). The spreading parameters were set to $\alpha = 0.5$ and $t = 0.7$ for brain (**A** and **B**) and $\alpha = 0.4$ and $t = 0.3$ for liver (**C** and **D**). These are the optimal parameters found by the between-dataset consistency analysis. Afterwards, we subset unmeasured genes which were recovered using network propagation and identified ageing-associated genes within these imputed genes. The median absolute propagated $\log_2$ fold change of the ageing-associated genes within imputed genes is higher than the median absolute propagated $\log_2$ fold change of all imputed genes. In the case of protein $\log_2$ fold changes in brain and liver smoothed with HD the difference is significant (one-sided Wilcoxon rank sum test; * $p \leq 0.05$). **E:** Number of differentially expressed genes between the more and less aggressive PCa tumors with HD at the mRNA layer before (upper left) and after multiple hypothesis testing correction (upper right) across the values of the spreading parameter. Below, distributions of the negative logarithm in base 10 of the t-test p-values for the different values of the spreading parameter before (lower left) and after multiple hypothesis testing correction (lower right) are depicted. The red lines have ordinate $-\log_{10}(0.05)$ and $-\log_{10}(0.1)$ respectively and correspond to the significance threshold. **F:** Same as in (**E**) but for RWR.

performed the same procedure on mRNA $\log_2$ fold changes (2,835 genes with missing transcript levels for brain and 3,222 for liver, 6,553 genes for brain and 6,166 for liver with measured transcript levels). To validate the imputed $\log_2$ fold changes, we compared $\log_2$ fold changes of known ageing-associated genes/proteins (taken from JenAge; http://agefactdb.jenage.de/, on December 4, 2017) to $\log_2$ fold changes of random genes/proteins. We expected that ageing-associated genes/proteins should on average have greater absolute $\log_2$ fold changes than random genes/proteins. This was indeed the case (Fig 6A–6D), suggesting that network propagation could be used to impute missing values and to identify new candidates of ageing-associated genes and proteins.

To further test the utility of network propagation for revealing molecular mechanisms, we performed differential expression (DE) analysis on the mRNA layer contrasting highly aggressive *versus* less aggressive PCa tumors for varying spreading parameters. Network propagation was performed for each sample independently. Thus, if the network propagation would just 'spread noise' within the samples, we would not expect increasing numbers of DE genes, which requires consistent differences between multiple samples. However, the number of DE genes increased with increasing network smoothing (Fig 6E and 6F) and a maximum number of DE genes was obtained prior to complete smoothing (i.e. before 'over smoothing'), especially when correcting for multiple testing. It is noteworthy that the majority of DE genes identified through this procedure are known cancer genes. Network propagation identified four additional DE genes (RHOC, LZTS1, ARHGDIG, PDIA2) after multiple testing correction by both algorithms (HD and RWR) at their optimum. Out of those four genes, three (RHOC, LZTS1, ARHGDIG) are known cancer genes [24–27]. This further supports the notion that network propagation enhances the signal of genes on relevant network regions while reducing the noise at the same time. Interestingly, the parameter values corresponding to the maximal number of DE genes were not the same as those from the MSE criterion. One potential explanation of this is that in order to maximize the number of genes differentiating highly from lowly aggressive tumors, more aggressive smoothing was needed to further decrease the gene variance.

## Discussion

Our investigation of two popular network propagation algorithms, Random Walk with Restart (RWR) and Heat Diffusion (HD) has revealed specific characteristics of the algorithms, it demonstrated potential artifacts resulting from the use of inappropriate network normalization methods and it serves as a guide for choosing optimal model parameters. Firstly, we have tested different graph normalization techniques for network propagation, which revealed that the degree normalized adjacency representation of the network does not induce a topology bias. On the other hand, the normalized Laplacian leads to the accumulation of higher scores on hub nodes with both

tested algorithms (HD and RWR). Graph normalization techniques inducing a topology bias might still be useful in practice though. However, one should be aware of a potential topology bias and about the impact of the chosen normalization approach on the final result.

Further, we have shown that optimizing the spreading parameters can be understood as a minimization problem of the bias-variance tradeoff, a well-known machine learning concept. We have assessed the impact of the spreading parameters on two multi-omics datasets with different approaches, using the bias-variance tradeoff as a first criterion, the transcriptome replicate consistency as a second and the mRNA and protein measurement consistency as a third criterion. The first two selection criteria require the existence of replicate measurements, which is a mild limitation as most studies will be performed in biological replicates. The last criterion is important in view of recent efforts for multi-omics data integration [14,28] and needs measurements at multiple omics layers. Here, the key requirement is that one requires congruency between the omics layers; i.e., it needs to be plausible that maximizing the correlation between the two layers actually increases the signal in the data. Interestingly, in the case of the rat liver data both intra- and inter-dataset consistency could not be further increased. In principle, two explanations exist for this observation: either the network used for the propagation was better suited for the brain data than for the liver data (S4 Fig), or the quality of the liver data was already at a limit that could not be further improved. The fact that both consistency measures were higher in liver compared to brain already before network propagation suggests that the latter explanation is more likely true. However, we wish to emphasize that the higher intrinsic consistency of the liver datasets may not only be due to technical reasons, but also due to a larger tissue homogeneity and smaller biological variability compared to the brain. For the prostate cancer dataset, more aggressive smoothing was needed for the protein compared to the mRNA layer in order to minimize the bias-variance tradeoff. This observation might reflect technical differences between protein and mRNA quantification or it might be caused by buffering of mRNA changes at the level of proteins [23]. Note that buffered fold changes would lead to a smaller increase in the bias$^2$ curve, which is what we observed in case of the PCa protein data (Fig 2I and 2J) compared to mRNA fold changes (Fig 2G and 2H). Additionally, the within-patient but also the between-layer similarity increased for most cases (patients or samples) with smoothing, while the few exceptions had low initial correlations. On the datasets tested here, using the MSE (first criterion) resulted in a more distinct optimum for the spreading parameters than using the correlation-based measures (second and third criterion).

The different criteria proposed for optimizing the spreading parameters might lead to distinct solutions. In this case, one needs to consider the research question and the context. For example, does one expect that the different omics layers should be consistent? Is it more important to increase the correlation between replicate measurements or between omics layers? Yet another possibility is to optimize a combination (e.g. a normalized sum) of the previous criteria so that they are met at the same time.

Analyzing the MSE improvement (propagation gain) for each gene separately revealed that the propagation gain of a gene depends on its variance, on its mean value, and on the neighbors' mean value.

Although mathematically we did not expect that RWR and HD necessarily lead to the same results, here we observed that with the empirical data at hand both methods lead to very similar conclusions. Interestingly, the correlation between log$_2$ fold changes propagated with HD and RWR is very high (S1 Fig). Although it is expected that RWR diffuses to larger regions, whereas HD remains more focused in a network sub-region [16,29], we observed a high similarity between the optimal results of both algorithms. This can be explained by the fact that the overall distances in the network we used were small (mean distance for the rat network: 4.5; for the

human network: 3.8) and we expect that differences between the algorithms would display more in networks with larger distances. Most biological networks have an average distance between 4 and 5 [30]. We assume that the more we know about these networks, the shorter the distances will become, because more knowledge usually means adding edges and only very rarely results in removing edges. As a consequence we expect that the effective differences between RWR and HD on biological networks will be small for many applications. However, the correlation starts dropping when the impact of network information, which is determined by the spreading parameters $t$ for HD and $\alpha$ for RWR, differs between the two propagation processes. Also, depending on the specific application and depending on the network used, differences between RWR and HD may be more relevant and should thus be considered.

Network propagation with use of optimal spreading parameters imputed missing values and uncovered known ageing-associated genes. Further, it was able to emphasize expression differences between tumors of different grades. These analyses confirmed the biological relevance of network smoothing.

To our knowledge, our study is the first to assess the impact of spreading parameters $\alpha$ and $t$ in RWR and HD systematically with several different approaches. Thus, our study serves as a template for how to identify optimal parameters, depending on the available data and tailored towards the research question.

## Material and methods

### Data used

For this study, the published young and old tissues transcriptome and proteome datasets from *Rattus norvegicus* were used [13]. The accession number for transcriptome is GEO: GSE66715 and for proteome ProteomeXchange: PXD002467. The data were derived from young (6 months) and old (24 months) Rat liver and brain samples with three biological replicates for each age. The transcriptome profiling was from the entire tissues whereas the proteomics measurements were from four subcellular fractions (nuc:nuclei, pn1:mitochondrial, pn2:cytoplasmic membrane, sol:soluble cytosolic proteins). Thus to collate the mRNA and protein abundance changes, the protein log fold changes from four subcellular fractions of a tissue were combined by calculating the weighted means.

Weighted mean $\log_2$ fold change$_{\text{protein } i}$ =

$$\sum\nolimits_{j=1}^{4} \frac{log_2 \; fold \; change_{protein \; i} \; in \; fraction \; j \cdot \#proteins \; in \; fraction \; j}{\#proteins \; in \; all \; fractions} \tag{9}$$

We also used the published transcriptome and proteome data of a multi-omics PCa study [14] consisting of 66 tumor samples belonging to four different grade groups (G1, G2, G3, G4/5) and 39 benign samples from 39 patients. For some of the patients (27 out of the 39), two tumor regions (TA1 and TA2) had been analyzed. Due to degradation issues for the mRNA layer, three tumor samples (corresponding to two patients) were removed from that layer. We quantified mRNA and protein $\log_2$ fold changes for the tumor samples (*versus* benign) as described in Charmpi *et al*. [14] resulting in matrices of dimension 14,281x63 and 2,371x66 respectively. Average (mRNA and protein) $\log_2$ fold changes were computed across all tumor samples available at the corresponding layer.

### STRING functional interaction network

The functional interaction network for Rattus norvegicus was retrieved from STRING database version 10.5 [8]. The generic network was filtered with the combined scores threshold of above

0.9 for high confidence interactions. This step of filtering was essential to avoid the flow of signal to the false positive edges. Additionally, to ensure convergence of the propagation process the largest connected network component was used. The filtered network had 9,388 nodes and 439,121 interactions, this network was employed for diffusing the transcriptome and proteome signals. In the case of PCa, we used the STRING interaction network for Homo sapiens (version 10). Applying the same filtering criteria, the resulting network consisted of 10,729 nodes and 118,647 edges.

## Random walk with restart

Network propagation with RWR was performed by using the R package BioNetSmooth, version 1.0.0 (https://github.com/beyergroup/BioNetSmooth.git). First the propagation starts by mapping the $\log_2$ fold changes of expression data or protein abundances on the protein-protein interaction network of *Rattus norvegicus*/*Homo Sapiens* with the function *network_mapping* by matching the gene IDs. Genes only present in the dataset but not in the corresponding network are removed. In the case that genes are present in the network but not have measured values, the *network_mapping* function replaces missing values with zero, so that the input vector and the interaction network matrix have the same dimension and gene order. Subsequently, the function *network_propagation* propagates the mapped $\log_2$ fold changes on the network according to Eq (1). The diagonal elements of $W$ are set to zero to avoid that the walker stays at the same node. The propagation runs iteratively with $i = [1, 2, 3, \ldots]$ until convergence. We used the following values for $\alpha$, $a \in \{0, 0.1, 0.2, 0.3, 0.4, 0.5, 0.6, 0.7, 0.8, 0.9, 0.93, 0.95, 0.98, 1\}$.

## Heat diffusion

Network propagation with HD was done according to Eq (2). Further, exponential of the network matrix multiplied by (-$t$) was computed using the *expm* function from Matrix R package, version 1.2–10. The propagated scores were obtained by multiplying the exponential with the $\log_2$ fold changes. We used the following values for $t$, $t \in \{0, 0.1, 0.2, 0.3, 0.4, 0.5, 0.6, 0.7, 0.8, 0.9, 1, 1.5, 2, 2.5, 3, 3.5, 4, 4.5, 5, 5.5, 10, 12, 15, 25, 50, 100, 1,000\}$.

## Graph normalization methods

Let $G = (V,E)$ be a simple, connected, undirected and unweighted graph where $V$ is the set of nodes and $E$ the set of edges. For the network matrix $W$, different graph normalizations can be employed. The simplest representation of a graph is its associated adjacency matrix $A = [a_{i,j}]$. The entries $a_{i,j}$ of the matrix are defined by

$$a_{i,j} = \begin{cases} 1, & \text{if } v_i \text{ is adjacent to } v_j \\ 0, & \text{otherwise,} \end{cases} \tag{10}$$

where $v_i$, $v_j \in V$. The adjacency matrix can be normalized row-wise by the degrees of the nodes. In this case, the entries are

$$w_D(i,j) = \begin{cases} \dfrac{-1}{d_i}, & \text{if } v_i \text{ is adjacent to } v_j \\ 0, & \text{otherwise,} \end{cases}$$

where $d_i$ denotes the degree of the vertex $v_i$. Additionally, the Laplacian of the graph can be used. It is defined by $W_L = D - A$, where $D$ is a diagonal matrix of the node degrees. The entries

of the matrix are filled by

$$w_L(i,j) = \begin{cases} d_i, & \text{if } i = j \\ -1, & \text{if } i \neq j \text{ and } v_i \text{ is adjacent to } v_j \\ 0, & \text{otherwise.} \end{cases}$$

The Laplacian can also be normalized by the degrees of the interacting nodes. In this case, the entries of the matrix are defined by

$$w_{\hat{L}}(i,j) = \begin{cases} 1, & \text{if } i = j \\ \dfrac{-1}{\sqrt{d_i d_j}}, & \text{if } i \neq j \text{ and } v_i \text{ is adjacent to } v_j \\ 0, & \text{otherwise.} \end{cases}$$

## Topology bias

Below we examine the presence or absence of a topology bias using different graph normalization and propagation approaches. In case of RWR, using the degree row-normalized adjacency matrix does not lead to a topology bias. The steady/limiting vector $F^a$ should satisfy Eq (1):

$$F^a = (1 - \alpha)F_0 - \alpha W_D F^a. \tag{11}$$

Denote by $\mathcal{I}$ the vector of all ones $(1, 1, \ldots 1)^T$. When $F_0 = \mathcal{I}$ then in order not to have topology bias, the steady vector $F^a$ should also be the vector $\mathcal{I}$ (definition of (no) topology bias), else stated the vector $\mathcal{I}$ should satisfy Eq (11). To show this, we take the right-hand side and get

$$(1 - \alpha)F_0 - \alpha W_D F^a = (1 - \alpha)\mathcal{I} - \alpha W_D \mathcal{I} = (1 - \alpha)\mathcal{I} + \alpha \mathcal{I} = \mathcal{I},$$

i.e. it does so for all $0 \leq \alpha \leq 1$. For the second equality, we have used that

$$-W_D \mathcal{I} = \begin{pmatrix} -\sum_j w_D(1,j) \\ -\sum_j w_D(2,j) \\ \vdots \end{pmatrix} = \mathcal{I}, \text{ since the row sums of } -W_D \text{ are all equal to 1.}$$

For the normalized Laplacian in case of RWR, the network matrix $W$ is set to $W_{\hat{L}D}$ to ensure that the diagonal elements of $W$ are zero. We will have similarly

$$(1 - \alpha)F_0 - \alpha W_{\hat{L}D} F^a = (1 - \alpha)\mathcal{I} - \alpha W_{\hat{L}D} \mathcal{I}$$

$$= (1 - \alpha)\mathcal{I} - \alpha \begin{pmatrix} \sum_j w_{\hat{L}D}(1,j) \\ \sum_j w_{\hat{L}D}(2,j) \\ \vdots \end{pmatrix} = \begin{pmatrix} -\alpha \sum_j w_{\hat{L}D}(1,j) + (1 - \alpha) \\ -\alpha \sum_j w_{\hat{L}D}(2,j) + (1 - \alpha) \\ \vdots \end{pmatrix}. \text{ But the row sums of}$$

the matrix with Laplacian normalization ($\sum_j w_{\hat{L}D}(i,j)$'s) are not equal, so the elements of the final vector above will be different. Thus the vector $\mathcal{I}$ does not satisfy Eq (11), consequently there will be some bias for $\alpha > 0$. The hub nodes being visited more often in the random walk, they are expected to gather higher scores compared to the non-hub nodes. In particular, when $\alpha = 1$ the limiting vector $F^a$ will be proportional to the vector of square roots of node degrees [31].

In case of HD, the Laplacian and degree row-normalized adjacency matrix do not lead to a topology bias. Firstly for the Laplacian, since for the matrix $W = W_L$ each row sums to 0, it

follows that each row of the matrix $e^{-W_L t}$ will be summing to 1 for all $t \geq 0$ [32]. As a consequence,

$$e^{-W_L t}\mathcal{I} = \mathcal{I}. \tag{12}$$

In other words, there will be no topology bias.

The degree row-normalized adjacency matrix $W_D$ is studied next. To avoid that for very large $t$ the algorithm may run into numerical instabilities, without loss of generality the adjacency matrix $W_{\hat{D}}(= W_D + I)$ was used instead to propagate with HD. We have:

$$e^{-W_{\hat{D}} t}\mathcal{I} = \mathcal{I}, \tag{13}$$

since the row sums of $W_{\hat{D}} = W_D + I$ are equal to 0 and thus the row sums of $e^{-W_{\hat{D}} t}$ will be all equal to 1 for each $t \geq 0$ like before. Hence, using the adjacency matrix $W_{\hat{D}}$ and the input vector filled with 1's, the output vector will be a vector with 1's and there will be no topology bias.

Eqs (11)–(13) also hold true in terms of expectation (since the operations involve linear combinations): if the population mean vector $\mu$ was the unit vector (i.e. all genes initially had the same mean value) then the propagated mean vector would also be the unit vector or in other words the bias curve would be constant and equal to 0.

For the normalized Laplacian $W = W_{\hat{L}}$, the row sums of the matrix $e^{-W_{\hat{L}} t}$ are not equal, consequently the elements of the propagated vector will be different for $t > 0$ and further depend on the node degree where hub nodes are expected to have higher scores since they are visited more often compared to the non-hub nodes like in RWR.

## Mean squared error computation

For each of the two tissues (brain and liver) and each replicate, we computed the MSE as the average (across the genes) squared difference between the smoothed replicate $\log_2$ fold changes for some value of the spreading parameter and the initial average $\log_2$ fold changes. This was done for all values of the spreading parameters using the two propagation algorithms (RWR and HD). Additionally, we computed the average (across the genes) squared difference between the smoothed replicate $\log_2$ fold changes and the corresponding smoothed average $\log_2$ fold changes (variance, see section below) as well as the average (across the genes) squared difference between the smoothed and initial average $\log_2$ fold changes (bias$^2$, see section below) with varying spreading parameters. For computing these quantities (MSE, bias$^2$ and variance) only measured network nodes were used. The latter two quantities (bias$^2$ and replicate variance) corresponding to the same value of the spreading parameter were also summed up. The previous computations resulted in six values for the MSE, variance and sum (corresponding to the six replicates) for each value of the spreading parameter (per tissue and propagation algorithm). These six values were log-transformed each time and their average and SD was computed in the transformed space. With these, a lower and an upper bound were computed as average-SD and average+SD respectively. Finally, the average, lower and upper bound were transformed back to the original space (i.e. the average will correspond to a geometric mean). The transformation was performed to avoid negative lower bounds (because average-SD could take a value below zero in the original space). The same approach was applied for the PCa study both on the mRNA and protein layer.

For the PCa study, we also generated MSE curves for each individual gene separately both at the mRNA and protein layer. More specifically, we computed for each measured network node its squared difference between the smoothed replicate $\log_2$ fold change for some value of the spreading parameter and the initial average $\log_2$ fold change. This was done for all values of the spreading parameters and all tumor samples using RWR. The previous computations

resulted in 66 values for the MSE of a gene (63 for the mRNA) for each value of $\alpha$. These values were log-transformed each time and their average was computed in the transformed space (to have the computations consistent with when computing an average (across the genes) MSE). Finally, the average was transformed back to the original space (i.e. the average will correspond to a geometric mean).

## Adding random noise to the liver data

We created a 'noisy' liver dataset as follows: for each gene $g$ we simulated six (independent) values from a normal distribution $\mathcal{N}(0, s_g^2)$ where $s_g^2$ corresponds to the variance of the $\log_2$ fold changes of that gene estimated across the six replicates, and subsequently added these simulated noise values to the original $\log_2$ fold changes. The noise values between genes were also drawn independently. Next, we performed network propagation (RWR and HD) on the noisy liver dataset and computed the MSE, bias$^2$ and variance as described in the above paragraph. As an estimate of the population mean vector $\mu$, we used the initial average $\log_2$ fold changes computed from the original liver dataset.

## The use of large spreading parameters leads to the loss of the biological signal

As the spreading coefficients increase we tend to lose the biological signal. With the degree row-normalized adjacency matrix, the variance between the node scores decreases (S3 Fig) and each input vector eventually converges to a vector with the same value for all elements as $\alpha$ approaches one and $t$ goes to infinity. Although computing the Pearson correlation between constant vectors is not possible, we were able to do so (numerically) due to the presence of small numerical fluctuations in the smoothed node scores at the end of propagation ($\alpha = 1$ for RWR and $t = 1,000$ for HD). However, the correlation results for large spreading parameters ($\alpha = 1$ and $t = 1,000$) are not (biologically) meaningful.

## Imputation of ageing-associated genes

After network propagation we subset unmeasured genes which were recovered using network propagation (n = 2,835 for RNA brain, n = 3,222 for RNA liver, n = 7,149 for protein brain, n = 7,318 for protein liver). Ageing-associated genes within these imputed genes based on the JenAge Ageing Factor Database [33], filtered for the organism *Rattus norvegicus*, were identified (n = 102 for RNA brain, n = 111 for RNA liver, n = 332 for protein brain, n = 349 for protein liver, see S1 Table). Additionally we tested if the median absolute $\log_2$ fold change of the ageing-associated genes within imputed genes is higher than the median absolute $\log_2$ fold change of the imputed genes using an unpaired one-sided Wilcoxon rank sum test. The p-values were adjusted using the Benjamini-Hochberg [34] correction approach.

## Differential expression analysis on the PCa study

For each of the network nodes measured on the mRNA layer and each value of the spreading parameter, we applied a t-test comparing its propagated (with each of the two propagation algorithms) mRNA $\log_2$ fold changes in the group G4/5 with those in the combined group G1-2 (consisting of the samples in the groups G1 and G2). To correct for multiple hypothesis testing, we used the Benjamini-Yekutieli method [35] (at each spreading parameter separately). The significance threshold was set to 0.05 and 0.1 before and after adjustment of the p-values respectively.

## Supporting information

**S1 Fig. High similarity between smoothed scores of RWR and HD.** The smoothed log fold changes for proteome of brain (**A**) and liver (**C**) tissue and transcriptome of brain (**B**) and liver (**D**) tissue computed with RWR (varying $\alpha$ between 0 and 1) and HD (varying $t$ between 0 and 1,000) are correlated. For the analysis, all genes which are present in the network (9,388 genes) are included.
(PDF)

**S2 Fig. Between-dataset consistency during propagation process in dependency of varying spreading parameters.** For brain tissue the correlation between mRNA and protein levels of ageing tissues during network propagation increases with both algorithms, HD (**A**) and RWR (**B**). Whereas for liver the consistency cannot be improved using network propagation with HD (**C**) or RWR (**D**). In the analysis, all genes which are present in the network as well as expressed and quantified with RNA-Sequencing and MS proteomics (n = 1,772 for brain, n = 1, 670 for liver) are included.
(PDF)

**S3 Fig. Effect of network smoothing on gene-gene correlation and within-sample variability.** A, B: The boxplots show the distributions of protein and RNA $\log_2$ fold changes for the brain (A) and liver (B) from Ori *et al*. [13] dataset before propagation ($\alpha = 0$, $t = 0$) and after propagation with RWR and HD using varying parameters ($\alpha = 0.5, 0.8, 1$ and $t = 1, 10, 1,000$). C, D: For each value of the spreading parameter, all pairwise correlations between genes across samples were computed using the corresponding propagated mRNA $\log_2$ fold changes of the PCa study with the two propagation algorithms (RWR and HD). Subsequently, the average pairwise correlation and standard deviation (SD; across the gene pairs) was computed for each value of the spreading parameter. Average pairwise correlation across the values of the spreading parameter with HD (C) and RWR (D) (blue curves). Error bars indicate the average +/- standard deviation. E, F: For each value of the spreading parameter, we computed for each PCa tumor sample the variance across its propagated mRNA $\log_2$ fold changes (within-sample variability) with the two propagation algorithms. Subsequently, the average within-sample variability and SD (across the tumor samples) was computed for each value of the spreading parameter. Average within-sample variability across the values of the spreading parameter with HD (E) and RWR (F) (blue curves). Error bars have been added in the same way. In the above analyses, we used the measured network nodes only to compute the pairwise correlations and the within-sample variability.
(PDF)

**S4 Fig. Bias induced through network propagation.** Observed $bias^2$ curves (blue) (as defined in *Material and Methods*) across the values of the spreading parameter for the brain (**A, B**) and liver (**C, D**) using HD (**A, C**) and RWR (**B, D**). We also applied the following randomization approach: we permuted the average $\log_2$ fold changes of the transcriptome for the brain and liver and computed $bias^2$ curves based on the permuted vectors. $Bias^2$ curves are shown for 10 permutations (orange). The computation of the $bias^2$ was based on the measured network nodes only. For the randomization approach, we only permuted the measured network nodes while the value of the unmeasured network nodes was always initially set to 0. **E:** Average $\log_2$ fold changes (of the measured network nodes) of the transcriptome for the brain (initial: dark blue, final (RWR, $\alpha = 1$): light blue) and liver (initial: dark red, final (RWR, $\alpha = 1$): pink) sorted in decreasing order.
(PDF)

**S5 Fig. MSEs and bias-variance tradeoff per grade group.** MSE curve (and corresponding bias-variance decomposition) across the values of the spreading parameter (similar to Fig 2) for each grade group separately: G1 (upper left), G2 (upper right), G3 (lower left) and G4/5 (lower right) for the mRNA with HD (**A**) and RWR (**B**) as well as for the protein layer with HD (**C**) and RWR (**D**).
(PDF)

**S6 Fig. A, B, C, D, E, F:** Same as Fig 4 but using Spearman correlation in the place of Pearson.
(PDF)

**S7 Fig. A, B, C, D, E, F:** Same as Fig 5 but using Spearman correlation in the place of Pearson.
(PDF)

**S1 Table. List of ageing-associated genes (STRING identifier and corresponding gene symbol) within imputed genes based on the JenAge Ageing Factor Database for each tissue (brain and liver) and each layer (mRNA and protein) in Ori *et al*. [13] dataset.**
(XLSX)

## Acknowledgments

We acknowledge Alessandro Ori (Fritz Lipmann Institute, Jena, Germany) for insightful discussions.

## Author Contributions

**Conceptualization:** Konstantina Charmpi, Andreas Beyer.

**Formal analysis:** Konstantina Charmpi, Manopriya Chokkalingam, Ronja Johnen.

**Funding acquisition:** Andreas Beyer.

**Methodology:** Andreas Beyer.

**Project administration:** Andreas Beyer.

**Resources:** Andreas Beyer.

**Software:** Konstantina Charmpi, Manopriya Chokkalingam, Ronja Johnen.

**Supervision:** Andreas Beyer.

**Visualization:** Konstantina Charmpi, Ronja Johnen.

**Writing – original draft:** Konstantina Charmpi, Manopriya Chokkalingam, Ronja Johnen, Andreas Beyer.

**Writing – review & editing:** Konstantina Charmpi, Ronja Johnen, Andreas Beyer.

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
