## [Decision Letter · Decision Letter 0]

10 Jul 2021

Dear Dr. Beyer,

Thank you very much for submitting your manuscript "Optimizing Network Propagation for Multi-Omics Data Integration" for consideration at PLOS Computational Biology.

As with all papers reviewed by the journal, your manuscript was reviewed by members of the editorial board and by several independent reviewers. In light of the reviews (below this email), we would like to invite the resubmission of a significantly-revised version that takes into account the reviewers' comments.

We cannot make any decision about publication until we have seen the revised manuscript and your response to the reviewers' comments. Your revised manuscript is also likely to be sent to reviewers for further evaluation.

Sincerely,

Teresa M. Przytycka

Associate Editor

PLOS Computational Biology

Jian Ma

Deputy Editor

PLOS Computational Biology

Reviewer's Responses to Questions

**Comments to the Authors:**

Reviewer #1: The paper is a pleasure to read. It studies the algorithm of network propagation and suggests methods to normalize its scores and optimize its parameters. It is well structured and provides all necessary algorithmic details.

However, it seems to neglect previous literature on this subject which is critical to its assessment.

Specifically, there have been several previous attempts for normalization of propagation scores to derive non-biased or even statistical scores. It is essential to compare the paper's results to those methods to determine the performance of the currently suggested method (row normalization). For a recent reference that reviews also earlier work in this regard see

"NetCore: a network propagation approach using node coreness", NAR 2020.

Additional comments:

- topology bias in my mind should be defined with respect to an empty prior which also coincides with setting alpha=1.

- In figure 1, can the authors compute the significance of the difference between hubs and non-hubs?

- it is mentioned that previous studies have chosen alpha arbitrarily but I don't think this is the case - rather performance for different alpha values was assessed.

- the limitations of the parameter selection method - e.g. having data with several replicates - should be clearly stated

Reviewer #2: Network diffusion approaches are commonly used for a variety of applications in biology – functional annotation, imputation, smoothing, etc. There are multiple parameterized approaches out there with little systematic comparison of various approaches and parameters. The authors take on this need and provide a systematic comparative analyses of network diffusion methods and parameters.

The work is well motivated, well executed and very well presented. The approaches seem sound. Of the various (matrix normalization) approaches, the results recommend against some of them based on topology bias (the result is biased by the network topology) one is still left with very little idea of which parameters to use for a specific application. Although this work is an honest exploration of a genuine challenge, I have several comments (hopefully to strengthen the work):

Major

1. Page 11: There are 6 young and 24 old mice. Why only 3 were used for inter-sample variation analysis?

2. Page 11: Authors have presented MSE across genes with various spreading parameter. It would be more informative to look at the distribution of errors across genes, and not just an overall MSE. And if there is a large variability in error for genes, what characterizes the genes with higher error v low error (something to do with network topology perhaps)?

3. Page 12: For the inter-replicate consistency, the improvement is minimal at best and in some cases achieved at alpha=1, which is not very useful. What is a biologist supposed to make of this? Some considered discussion is needed at the minimum.

4. Fig 5A-D: This is nice, but please quantify what fraction of such genes/proteins with missing value were deemed significantly differential based on imputed value.

5. Fig 5E: Is it correct that ONLY 4 genes are differential after multiple testing correction? Any comments on this? If so, is this even a good data to be analyzing?

6. Overall, the results are quite mixed in terms of showing the value of network propagation. Are there range of parameter values that can be recommended that covers the optimal choice across various applications and various optimality criteria? Without this, what is the impact of this work for actual application of these methods?

7. Even though, based on topological bias, which manifests itself only at the extreme levels of alpha value, authors have ruled out certain smoothing approaches (which by the way has been used a lot), it will still be useful to include those methods in the evaluations – it may still be useful for certain applications.

8. Finally, given that there is SO MUCH data out there, the application seems very narrow. I would like to see somewhat broader benchmarking, especially using datasets where the key genes are established.

Minor

9. Supplementary Fig 3: Y-axis should be labelled ‘inter-sample..’, not ‘within-sample..’

10. Page 12: wherever correlations are used, please use Spearman correlation to avoid outlier effect, and specify.

**Have the authors made all data and (if applicable) computational code underlying the findings in their manuscript fully available?**

Reviewer #1: None

Reviewer #2: Yes

PLOS authors have the option to publish the peer review history of their article (what does this mean?). If published, this will include your full peer review and any attached files.

Reviewer #1: No

Reviewer #2: No
---

## [Decision Letter · Decision Letter 1]

12 Oct 2021

Dear Dr. Beyer,

We are pleased to inform you that your manuscript 'Optimizing Network Propagation for Multi-Omics Data Integration' has been provisionally accepted for publication in PLOS Computational Biology.

Best regards,

Teresa M. Przytycka

Associate Editor

PLOS Computational Biology

Jian Ma

Deputy Editor

PLOS Computational Biology

Reviewer's Responses to Questions

**Comments to the Authors:**

Reviewer #1: I am satisfied with the revision and recommend publication.

Reviewer #2: I am satisfied with the authors' response to my initial comments.

**Have the authors made all data and (if applicable) computational code underlying the findings in their manuscript fully available?**

Reviewer #1: Yes

Reviewer #2: None

PLOS authors have the option to publish the peer review history of their article (what does this mean?). If published, this will include your full peer review and any attached files.

Reviewer #1: No

Reviewer #2: No

---

## [Editor Report · Acceptance letter]

5 Nov 2021

PCOMPBIOL-D-21-00994R1 

Optimizing Network Propagation for Multi-Omics Data Integration

Dear Dr Beyer,

I am pleased to inform you that your manuscript has been formally accepted for publication in PLOS Computational Biology. Your manuscript is now with our production department and you will be notified of the publication date in due course.

With kind regards,

Katalin Szabo
